# Solid and Secure Multi-Party Computation for a Circular Economy

Jan Grau[1,2,*,†], Aliza Maftun[3,†], Kimberly Garcia[2,†] and Simon Mayer[2]

[1]*Empa, Lerchenfeldstrasse 5, St. Gallen, 9000, Switzerland*

[2]*University of St. Gallen, Rosenbergstrasse 30, St. Gallen, 9000, Switzerland*

[3]*Siemens Foundational Technologies, 755 College Rd E, Princeton, NJ 08540*

## Abstract

Advancing the circular economy requires accessible, high-quality data from across supply chains. While platforms such the ISO-compliant *Green Design for Tomorrow (GDT)* already offer services to support circular transformation, much of the data needed for effective circular decision-making remains locked within corporate IT systems such as Enterprise Resource Planning systems. This data is not only difficult to extract due to deep integration with core business processes and controlled APIs, but companies are also reluctant to share it for fear of losing competitive advantage: the data holds business value. In this paper, we demonstrate the use of a ontology-based architecture and the Solid protocol to address these challenges. By making each member of the supply chain a data provider based on the Solid protocol and performing Life Cycle Assessment in a privacy-preserving manner through Secure Multi-Party Computation, we enable cross-supply-chain data integration while companies retain control of their data and its usage. This supports transitioning to a circular economy while offering a path to solve the involved confidentiality-transparency conundrum.

## Keywords

Social Linked Data, industrial ecology, carbon footprint, Life Cycle Analysis, WebAssembly Components

## 1. Introduction

Applications that support the circular economy, whether for creating Environmental Product Declarations (EPDs), tracking material provenance, or optimizing reuse and recycling, share a common requirement: they need comprehensive data spanning entire supply chains. The *Green Design for Tomorrow (GDT)*[1] platform offers such services by facilitating sustainable product design and enabling companies to create certified EPDs based on data from their supply chain partners.

Taking a broader perspective, Life Cycle Assessment (LCA) provides the methodological foundation for quantifying environmental impacts, like greenhouse gas emissions, and is therefore an important tool for the circular economy. A major challenge in LCA is the *Scope 3 Problem*. Scope 3 emissions are indirect emissions occurring across a company's value chain—from suppliers, logistics providers, and downstream users—that are outside the company's direct operational control. These emissions exceed operational emissions by a factor of 26 on average, reaching ratios as high as 92:1 in the retail sector, yet in 2023 only 15% of CDP-disclosing companies have established upstream Scope 3 targets [1]. This disparity is inherent to Scope 3 itself: organizations fundamentally lack both, visibility and leverage, to measure emissions they do not control. Hence, an ideal solution would allow each company reporting only its direct *Scope 1* emissions—greenhouse gas emissions from sources that a company owns or directly controls—within a shared, interlinked data infrastructure, such that one company's Scope 1

*Solid Symposium'26: THE 4TH SOLID SYMPOSIUM, April 30 - May 1, 2026, London, UK*

*Corresponding author.

†These authors contributed equally.

✉ jan.grau@empa.ch (J. Grau); aliza.maftun@siemens.com (A. Maftun); kimberly.garcia@unisg.ch (K. Garcia); simon.mayer@unisg.ch (S. Mayer)

🆔 0009-0006-0565-2034 (J. Grau); 0009-0006-6834-1655 (A. Maftun); 0000-0002-4971-2944 (K. Garcia); 0000-0001-6367-3454 (S. Mayer)

[1]See https://greendesigntomorrow.com/ (last accessed: 13.02.2026)

emissions automatically contribute to another's Scope 3, thereby transforming an intractable accounting challenge into a collaborative, automatable process.

The data relevant for this accounting resides within corporate IT systems and is generated as a byproduct of core business processes and usually deeply embedded in the enterprise architecture, e.g., in Enterprise Resource Planning (ERP) systems, making it costly and complex to extract or restructure. Even if it could be readily extracted, a more fundamental tension is that sharing this data with external stakeholders is typically perceived as a threat to competitive advantage. This tension could be resolved by providing *privacy-preserving* access to the data, where companies retain their data as a single source of truth within their own systems while external platforms access it in a controlled, privacy-preserving way. This assures that sensitive business information, such as production volumes, supplier relationships, or process efficiencies, is neither disclosed to upstream or downstream entities, nor the public. Techniques such as differential privacy, fully homomorphic encryption, and Secure Multi-Party Computation (SMPC) allow multiple parties to compute aggregated results without revealing their individual inputs. Of these, we consider SMPC to be best suited to LCA computation: it enables decentralized emissions calculations across a supply chain without exposing any participant's data to a single server. Fully homomorphic encryption, while theoretically appealing, currently proved to be too slow for practical use, and differential privacy lacks the structural sophistication that LCA demands.

Finally, as the volume of data produced across supply chains continues to grow, questions of provenance and ownership become increasingly important [2]. Companies need assurance that they retain control over who accesses their data, under what conditions, and for what purposes. The Solid Protocol addresses these requirements through a fine-grained access control system that places data ownership firmly in the hands of the data producer [3, 4]. As Solid is built on open standards and is designed to be easily self-hosted, it allows companies to deploy it as a natural extension of their existing IT infrastructure rather than as a dependency on an external platform [4, 5]. This combination of interoperability, sovereignty, and ease of integration makes Solid a compelling foundation for decentralized data sharing in the circular economy [6, 5].

## 2. Approach: Solid and SMPC to overcome the Confidentiality-Transparency Conundrum in Supply Chains

Drawing on the capabilities and challenges outlined above, we present an ontology-based Virtual Pod Provider approach and apply this to GDT. As a concrete use case, we demonstrate privacy-preserving LCA calculation using SMPC based on Shamir secret sharing, illustrating how Scope 3 emissions can be computed collaboratively across a supply chain while keeping each participant's sensitive data confidential.

Our approach is structured into three layers. The first layer is the LCA and Ecodesign application i.e., GDT, which hosts its existing services and provides each company with a sandboxed, multi-tenant application environment. This application layer is extended as part of this project: since GDT already manages relevant data, it can serve both as a data source and as an actor within the broader system. Specifically, GDT communicates with the SMPC system layer, which in turn interfaces with the Solid layer. The Solid layer extends the Solid Community Server, which implements the Solid Protocol and serves as the decentralized data access point with the plugin interface extension described below. In the following subsections, each of these layers is explained in detail.

### 2.1. Green Design for Tomorrow

The GDT is a platform for LCA and Ecodesign that calculates among other data with Bills of Materials from companies and offers a range of services relevant to the circular transformation. For the purposes of this project, we focus specifically on the LCA calculation service it provides. This service is extended here to integrate with the SMPC system layer described in the following subsection, enabling privacy-preserving LCA computation across organizational boundaries.

## 2.2. SMPC System Layer

Applications such as GDT will interact with the core SMPC system layer for the privacy-preserving calcuation. In order to perform the calculation in a privacy-preserving way, the SMPC system requires at least three computation servers to perform the core LCA calculation using Shamir Secret Sharing. The calculation follows the standard LCA calculation formula ($QBA^{-1}f$) [2], operating entirely on secret-shared values. Any server can act as an SMPC participant, and as long as the servers do not collude [3], none of them can determine the actual values being processed. This setup also allows the number of computation parties to be scaled to more servers to further strengthen privacy guarantees. However, since the application is grounded in a core LCA ontology (ORIONT) [9] and the calculation logic is implemented generically, any Solid-compatible server could be used.

## 2.3. Solid at its Core

The SMPC layer depends on access to distributed lifecycle data. The Solid protocol, building on the Linked Data Platform (LDP) [10], provides a natural solution to the Scope 3 problem introduced earlier. Each company maintains its own product and process data in its own data store, linking to external companies for external inputs or outputs. The SMPC layer can then perform the LCA calculation over this federated data without revealing any company's individual contributions. However, given the data residency constraints outlined earlier, the data required for the calculations should remain within each company's core IT system. To address this, a plugin system layer was developed using the WebAssembly (WASM) Component Model [11]. The WASM component model provides secure sandboxing of features and computations, which is leveraged here to ensure that plugins operate within clearly defined boundaries [12]. To ensure easily automatable SMPC execution, the authors opted to provision data directly through the plugin system, rather than employing a separate Data Provisioning Proxy (DPP) as proposed by Both et al. [2] Each plugin exposes a well-defined export interface, specified in WIT (WebAssembly Interface Type) terms, consisting of three functions to describe and execute the required transformations. Some transformations, however, require access to external data, such as a bridge ontology or a specific source file. For this purpose, an import interface is defined through which the host provides a `fetch-resource` function to the plugin. The plugin can only access data through this interface, which ensures that the hosts' authentication system remains in control. The plugin thus operates in a secure sandbox: it can transform data and retrieve authorized resources, but nothing more. Because the plugin interface supports a variety of input sources, applications such as GDT can serve a dual role: not only initiating SMPC-based calculations, but also contributing their own data to the computation in privacy-preserving way. Two plugin implementations are provided for this application: the Bridge Ontology based plugin, and the RML plugin.

**Bridge Ontology Based Plugin** The Bridge Ontology Based Plugin is built on an extended version of json2rdf[4] and xml2rdf[5], general-purpose JSON/XML-to-RDF mappers adapted for the WASM Component Model [11]. These tools transform any JSON or XML document into a generic RDF structure. Based on this intermediate representation, a bridge ontology can be written, either manually or with the assistance of an LLM, to map the generic RDF to the desired target ontology. The mapping is then executed by a SPARQL-based core engine built on Nemo [13], a Datalog rule engine that supports OWL-based rules, and on SHACL Advanced Features (SHACL-AF) [14]. SHACL-AF support is necessary because certain source data models require the creation of new nodes in order to conform to the target ontology.

---

[2] Where $\mathbf{Q}$ = characterization matrix, $\mathbf{B}$ = biosphere matrix, $\mathbf{A}$ = technosphere matrix, and $\mathbf{f}$ = final demand vector; see [7].

[3] Although Shamir Secret Sharing with parameters $(t, n)$ would ensure that up to $t - 1$ parties could theoretically collude without learning the secret, where $t$ is the reconstruction threshold and $n$ the total number of parties [8].

[4] https://crates.io/crates/json2rdf

[5] https://crates.io/crates/xml2rdf

**RML Plugin**    De Mulder et al.[5] have demonstrated a functioning RML-based approach for circular data sharing within Solid. Building on this, the authors developed an RML plugin based on mappingloomrs[15]. The WebAssembly-based architecture additionally enables deployment in browser-only environments. Future extensions to support direct database access are planned.

Beyond the Plugin System, Solid's fine-grained access control enables an even deeper integration with the SMPC architecture. Specifically, companies can perform the share generation step of Shamir Secret Sharing entirely within their own Solid-controlled environment. Using Solid's authentication and authorization mechanisms, each resulting share can then be selectively disclosed to a different SMPC computation server. For example, Server A may be granted access only to Share 1, while Server B receives access only to Share 2. In this way, the core security guarantees of Secure Multi-Party Computation, namely that no single server ever sees the complete data, are enforced directly at the data access layer rather than relying solely on the computation protocol itself.

## 3. System

To demonstrate the system, we constructed a representative supply chain, enabling the generation of diverse yet realistic product supply chains. These supply chains were integrated into the computation process alongside existing nodes from GDT. Supply chain members that do not yet have a GDT integration were simulated using the SAP Cloud Application Programming Model (SAP CAP) [16] to replicate realistic ERP API interfaces. For the demonstration system, a Docker-based setup was chosen in which each participant—including all involved companies, Empa, and the auditors—is provided with a dedicated Solid Community Server [17] instance, extended with the plugin interface described above. The Solid Community Server [17] was chosen for this project due to its flexibility and its own component-based architecture, which allows a virtual resource provider to be added with relative ease. Its flexible configuration system enables adaptation to diverse company landscapes and hosting requirements. The SMPC calculations were implemented using the Jiff library [6], and the user interface was built with Next.js [7]. Figure 1 depicts the architectural setup of a potential four member supply chain application. A more detailed workflow demonstration through a calculation can be found in the presentation video.

---

[6]https://github.com/multiparty/jiff
[7]https://nextjs.org/

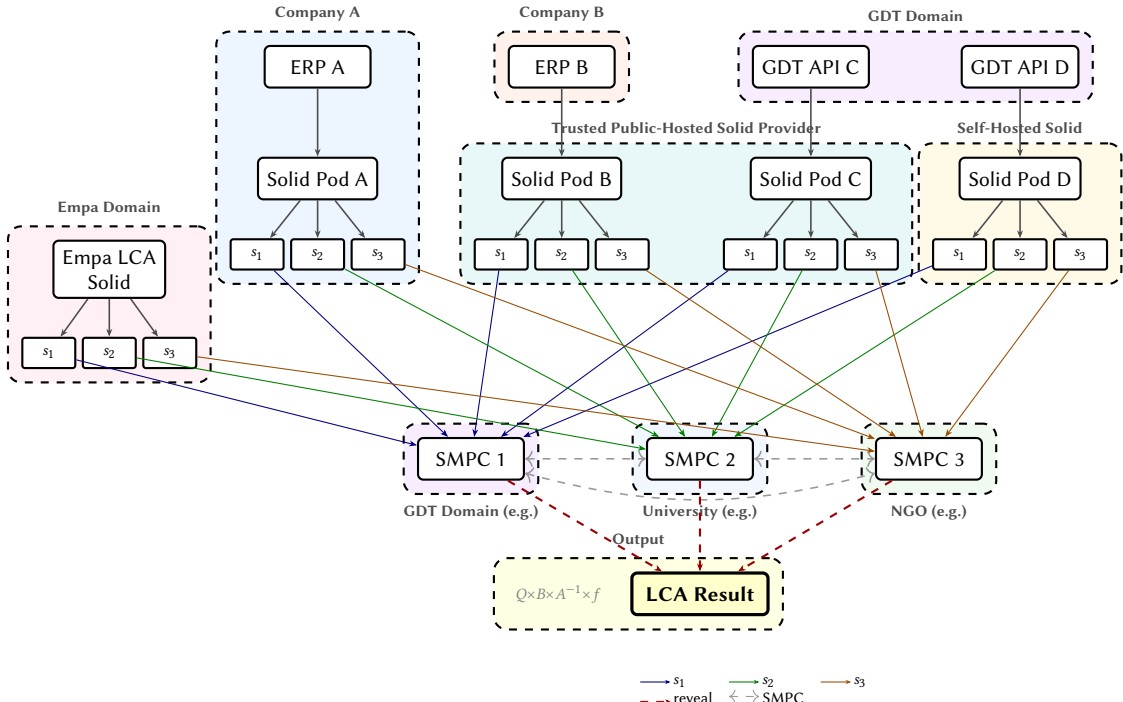

**Figure 1:** System overview of a potential four member supply chain setup demonstrating SMPC LCA calculation, where Empa acts as a background LCA database.

## 4. Conclusion

In this paper, we presented an ontology-based Virtual Pod Provider system for application such as the ISO compliant GDT that combines the Solid protocol, WASM-based plugins, and Secure Multi-Party Computation to enable privacy-preserving Life Cycle Assessment across supply chains. The architecture demonstrates that meaningful circular economy data exchange is possible without requiring companies to expose sensitive business information. By grounding the system in the ORIONT ontology and leveraging the plugin interface for flexible data transformation, the approach accommodates diverse corporate IT landscapes while maintaining semantic interoperability. Several directions for future work remain. First, the integration of authentication mechanisms for JSON-based APIs needs to be formalized, as the current setup assumes internal hosting. Second, extending the RML plugin to support direct SQL database access would broaden the range of enterprise data sources that can be connected. Third, additional plugins could further lower the barrier to onboarding new supply chain participants. Fourth, despite being faster than homomorphic encryption, the SMPC calculation speed remains a significant bottleneck for practical adoption, but the authors are actively developing an optimized approach to address this limitation. Finally, as large language models continue to mature, their potential to reduce the cost of ontology mapping warrants further investigation, complementing rather than replacing the structured, semantically grounded approach presented here.

## Acknowledgments

We would like to thank our collaborators at Siemens AG Rolph Kreis, Dr. Martin Wimmer and Regine Castilla for their support.

## Declaration on Generative AI

During the preparation of this work, the author(s) used Claude Sonnet 4.5 and Claude Haiku 4.5 in order to: Grammar and spelling check. After using these tool(s)/service(s), the author(s) reviewed and edited the content as needed and take(s) full responsibility for the publication's content.

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
