# OpenReview forum: "Solid and Secure Multi-Party Computation for a Circular Economy"
_SolidProject.org/SoSy/2026/Privacy_Session — SoSy2026-Privacy Paper_

### Official Review · ~Rui_Zhao15 · 2026-03-02
**Sensible alternative exploration of SMPC with Solid**

**Rating:** 7
**Confidence:** 5

**Review:**

This paper presents an idea / system of running SMPC with Solid to address the confidentiality considerations of different parties wanting to collaborate on a computation. It features a pre-calculation of secret shares of data as concrete data resources in the Pod, and apply access controls over these resources, as a way to reduce the introduction of additional components for that purpose (for SMPC; contrary to existing approaches, such as [Libertas](https://arxiv.org/abs/2309.16365)). The authors also briefly discussed some additional components as the "plugins" for external data integration/fetching.
In general, the paper presents an interesting idea of running SMPC with existing Solid infrastructure, theoretically under a sensible resource cost. I would support it being included in the session.

In the meantime, there are some limitations of this paper that I want to highlight, both to the chairs and to the authors:

1. It only discussed a system, but not giving any proofs of its existence -- no code, no benchmark, etc.
2. The connection between SMPC and the "plugin" part is quite vague. I find it hard to understand how the plugins contributed to or made it more difficult to running SMPC.
3. I did not find discussions on the trust / security relationships between the proposed SMPC parties. It would be useful to justify the security properties of those parties (e.g. why that holds?).
4. There are existing explorations of combining SMPC with Solid, especially [Libertas](https://arxiv.org/abs/2309.16365) (and [its optimization](https://arxiv.org/abs/2310.20062)). Maybe the authors could make it clear about the difference between theirs and other works.

The paper would also benefit from cleaning up the introductory parts, as it was hard to grasp the main problem that the paper aims to solve without going deep into further sections.

(p.s. My scores are based on the generous assumption that the system does exist, and my considerations can be easily addressed -- addressing them should be relatively easy, at least to an acceptable state.)

---

### Official Review · ~Víctor_Rodríguez-Doncel1 · 2026-03-05
**Looks nice (but I am not a true expert)**

**Rating:** 6
**Confidence:** 2

**Review:**

This architecture combines Solid's fine‑grained data sovereignty with Secure Multi Party Computations's privacy‑preserving computation and the WASM Component Model plugin layer. I am not an expert in WASM and I don't feel confident reviewing that part.
Apparently, the paper's solution enables companies to share only what is necessary while keeping sensitive ERP data fully under their control and not exposing competetive information.

As an ontology expert, I paid particular attention to the "ontology" in the ontology-based Virtual Pod Provider. Well, the referred paper Orienting’s LCA methodology–lessons learnt from industrial testing doesn't explain it well enough, and I was not able to inspect the ontology. (the namespace seems to be https://orienting.eu/oriont#, but it does not resolve...). I did not find any evidence sustaining the claim of the "ontology-based". Probably there are, but they should be shown...

I see other problems. Not much evaluation, no demo, no github repo, no real data described. But I guess that for a symposium this is not a critical problem.

Finally, I have the impression that other works should have been commented (differences, etc.) like Rui Zhao's "Libertas" (2023) https://export.arxiv.org/abs/2309.16365...

---

### Decision · Program_Chairs · 2026-03-09

Accept (Paper)